# Anti-Inflammatory Responses Produced with *Nippostrongylus brasiliensis*-Derived Uridine via the Mitochondrial ATP-Sensitive Potassium Channel and Its Anti-Atherosclerosis Effect in an Apolipoprotein E Gene Knockout Mouse Model

**DOI:** 10.3390/biom14060672

**Published:** 2024-06-08

**Authors:** Yingshu Zhang, Xin Ding, Caiyi Yuan, Yougui Yang, Qiang Zhang, Jiakai Yao, Ying Zhang, Junhong Wang, Yang Dai

**Affiliations:** 1School of Public Health, Nanjing Medical University, Longmian Avenue 101, Nanjing 211166, China; 2Jiangsu Provincial Medical Key Laboratory, Jiangsu Provincial Key Laboratory on Parasite and Vector Control Technology, National Health Commission Key Laboratory of Parasitic Disease Control and Prevention, Jiangsu Institute of Parasitic Diseases, Yang Xiang 117, Wuxi 214064, China; 3Department of Parasitic Disease Control, Lishui District Center for Disease Control and Prevention, Middle Street 17, Nanjing 211200, China; 4Department of Cardiology, the First Affiliated Hospital of Nanjing Medical University, Guangzhou Road 300, Nanjing 210029, China

**Keywords:** *Nippostrongylus brasiliensis*, uridine, anti-atherosclerosis, mitochondrial ATP-sensitive potassium channel

## Abstract

Atherosclerosis (AS) has become the leading cause of cardiovascular disease worldwide. Our previous study had observed that *Nippostrongylus brasiliensis* (Nb) infection or its derived products could inhibit AS development by inducing an anti-inflammatory response. We performed a metabolic analysis to screen Nb-derived metabolites with anti-inflammation activity and evaluated the AS-prevention effect. We observed that the metabolite uridine had higher expression levels in mice infected with the Nb and ES (excretory–secretory) products and could be selected as a key metabolite. ES and uridine interventions could reduce the pro-inflammatory responses and increase the anti-inflammatory responses in vitro and in vivo. The apolipoprotein E gene knockout (ApoE^−/−^) mice were fed with a high-fat diet for the AS modeling. Following the in vivo intervention, ES products or uridine significantly reduced serum and liver lipid levels, alleviated the formation of atherosclerosis, and reduced the pro-inflammatory responses in serum or plaques, while the anti-inflammatory responses showed opposite trends. After blocking with 5-HD (5-hydroxydecanoate sodium) in vitro, the mRNA levels of M2 markers were significantly reduced. When blocked with 5-HD in vivo, the degree of atherosclerosis was worsened, the pro-inflammatory responses were increased compared to the uridine group, while the anti-inflammatory responses decreased accordingly. Uridine, a key metabolite from *Nippostrongylus brasiliensis*, showed anti-inflammatory and anti-atherosclerotic effects in vitro and in vivo, which depend on the activation of the mitochondrial ATP-sensitive potassium channel.

## 1. Introduction

Atherosclerosis (AS), characterized by low-grade chronic inflammation in the intima of the arterial wall caused by the subendothelial retention of plasma-derived apolipoprotein B [1,2,3], has become the leading cause of cardiovascular disease worldwide, including ischemic heart disease, ischemic stroke, and peripheral artery disease [4]. Low-grade chronic inflammation caused by the infiltrated multiple inflammatory cell subsets, including macrophages, T lymphocytes, mast cells, etc., is an important pathophysiological process in the initiation and development of AS [5]. Among these inflammatory cell subsets, macrophages have been shown to be most important as they may be involved in a number of AS pathological processes such as lipid deposition, foam cell formation, and the rupture of AS plaques, making them an important target for AS interventions [6]. During the process of AS formation, classical activated macrophages (CAM, M1) showed pro-inflammatory effects, which could promote the formation AS plaques and increase plaque instability, while alternative active macrophages (AAM, M2) could play an anti-inflammatory role, which could maintain AS plaque stability and promote AS plaque regression [7]. Previous studies had shown interventions, targeting specific miRNAs (miR-155, miR-147, miR-21, etc.) or transcription factors (STAT-6, PPARγ, NF-κB, etc.), could specifically inhibit the activation of M1 macrophages and promote the activation of M2 macrophages, which could effectively inhibit the formation of AS [8]. Therefore, the effective induction of anti-inflammatory responses, especially the activation of M2 macrophages, have become an important means of inhibiting the development of AS.

Epidemiological evidence showed a significant negative association between human helminth infections and the prevalence of inflammation-related diseases [9]. Building on this, Strachan proposed the concept of the “hygiene hypothesis” as early as 1989, which assumes that this is accompanied by the continuous decline in infections such as parasitic infections. The possibility of exposure or infection to pathogens in human childhood gradually decreased, which accordingly affected the normal training and development of the body’s immune system and further increased the morbidity of inflammation-related diseases in adulthood [10,11]. Based on this hypothesis, numerous studies have reported the use of helminth infections or helminth-derived molecules for intervention in inflammatory diseases such as allergies, asthma, and metabolic disorders in the course of the disease, which is manifested in symptom improvement, weight loss, a reduction in BMI index, a reduction in total cholesterol levels, and the alleviation of insulin resistance [12,13,14]. This may suggest that helminth infection or its derived molecules could have good pharmacological effects on the occurrence and development of inflammatory diseases, providing a new idea for disease prevention and treatment.

*Nippostrongylus brasiliensis* (*N. brasiliensis*, Nb), also known as the rat hookworm, has often been used as a model for human hookworm studies due to its similar life cycle and induced immune response [15,16]. Previous studies have shown that infection with *N. brasiliensis* can induce a high level of type II immune responses, such as IL-4, IL-10, and IL-13 secretion, and activate innate lymphoid cell type 2 (ILC2) and M2 polarization, all with anti-inflammatory effects [17,18]. Our previous study has also shown that *N. brasiliensis*-derived products can stimulate macrophages with M2 polarization in vitro [19]. Furthermore, in the AS model (ApoE^−/−^ mouse on high-fat diet), *N. brasiliensis* infection or its derived products could induce an anti-inflammatory state and effectively inhibit the occurrence and development of AS, as evidenced by reduced aortic arch plaque areas and liver lipid contents, downregulated serum low-density lipoprotein (LDL), and upregulated anti-inflammatory cytokine levels (IL-10 and IL-4) manifested in serum [20]. However, interventions with direct helminth infection or its derived mixed products could have several concerns including ethical constraints, safety considerations, and not guaranteed effects. Therefore, screening and evaluation for helminth-derived molecules with anti-inflammatory or anti-atherosclerotic effects might be more suitable and feasible for the following inflammatory-related disease intervention.

Previous studies had reported that successfully identified molecules (proteins or peptides) derived from hookworms, such as AceES-2, Ac-AIP-2, and Na-AIP-1, had shown good anti-inflammatory activities in vitro and in vivo, with an ideal palliative effect in inflammation-related diseases [21]. The host intestinal parasitism of adult *N. brasiliensis* worms and excretory–secretory (ES) products, including proteins, carbohydrates, and small molecular metabolites, form the “first-line molecules” in parasite–host interactions, which represent an ideal resource pool for the anti-inflammatory molecules mining from parasites [22]. Among the excretory–secretory (ES) products of *N. brasiliensis*, small molecule metabolites attract more attention in the following translational drug development study due to their low antigenicity and good drug availability. Encouragingly, some studies had suggested that helminth-derived metabolites showed potential pharmacological activity [23], which may improve the feasibility of screening active molecules based on helminth-derived metabolites.

In the present study, we performed a metabolic analysis to screen *N. brasiliensis*-derived metabolites with the anti-inflammatory effects of ES products and *N. brasiliensis*-infected mice and identified their anti-inflammatory effects using a lipopolysaccharide (LPS)-induced acute inflammation model and bone marrow-derived macrophage model. Furthermore, we investigated the anti-atherosclerosis effect of the metabolite using ApoE^−/−^ mouse models and elucidated its possible functional mechanism.

## 2. Materials and Methods

### 2.1. Ethical Statement

The present study conformed to the Declaration of Helsinki and was approved by the Ethics Committee of the Jiangsu Institute of Parasitic Diseases (accession numbers: JIPD-2020-007 and JIPD-2021-002). All animals used in this study were raised in the animal center of the Jiangsu Institute of Parasitic Diseases (JIPD) in a standard environment with a 21–25 °C temperature, 65% humidity, 10–14 h light–dark cycle, and free eating and drinking.

### 2.2. Nippostrongylus Brasiliensis and Excretory–Secretory Products (ES)

As a gift from the laboratory of Professor Alex Loukas at James Cook University in Australia, the *N. brasiliensis* parasite was maintained and cycled in our laboratory for approximately six years. The parasite was introduced into Sprague–Dawley rats (male, 300 g) provided by the Animal Center of the Jiangsu Institute of Parasitic Diseases (JIPD, Wuxi, China) according to the previously published protocol [15]. *N. brasiliensis* adult worms (L5 stage) were collected from infected rats on day 7 post-infection and used to prepare ES products according to our previous report [24]. Briefly, adult worms were collected from the small intestine of infected rats using serial methods, including washing with Dulbecco’s phosphate-buffered saline (DPBS) (Gibco-Thermo Fisher, Waltham, MA, USA), incubated for 2 h at 37 °C, and separated with debris of the host. After washing twice with DPBS containing antibiotic and antimycotic drugs (Gibco-Thermo Fisher, Waltham, MA, USA), the adult worms were counted and placed in 24-well plates (500 worms/well) for the next 7 days of culture in RPMI-1640-transferred medium (Gibco-Thermo Fisher, Waltham, MA, USA) with antibiotic–antimycotic drugs (1×) under 37 °C and 5% CO_2_ conditions. The culture supernatant collected every 24 h was centrifuged (at 2000× *g* and 4 °C) for 10 min to remove the eggs and parasite fragments and filtered with a 0.22 μm filter to obtain the final ES. After concentration determination, the ES products were stored at −80 °C and used for the subsequent experiments.

### 2.3. Metabolomics Analysis

Two types of samples were prepared for metabolomics analysis, including intestinal content samples from *N. brasiliensis*-infected mice and ES product samples from adult worms (L5), which were prepared according to the above method. To prepare intestinal contents, ten male C57BL/6 mice aged 5 to 6 weeks were randomly divided into two groups, including the infected group and the control group, with 5 mice in each group. In the infected group, each mouse was infected with 250 third instar (L3) larvae of *N. brasiliensis* by subcutaneous injection in the back. Twelve days after infection, all mice were euthanized by carbon dioxide asphyxiation. The intestine was separated and the contents were scraped into sterile 1.5 mL tubes, weighed, and stored at −80 °C until use. All samples, including intestinal content samples (200 mg each) from infected (I, *n* = 5) and control (C, *n* = 5) mice, ES product samples (200 μL each, ES, *n* = 3), and control samples (B, *n* = 5) were used for further metabolomics analysis.

After metabolite extraction of all samples, chromatographic separation and LC-MS detection were carried out stepwise. Chromatographic separation was performed in a Thermo Ultimate 3000 system equipped with an ACQUITY UPLC^®^ HSS T3 column (150 × 2.1 mm, 1.8 μm, Waters, Milford, MA, USA) and maintained at 40 °C. The autosampler temperature was 8 °C. The gradient elution of the analytes was carried out with 0.1% formic acid in water (A) and 0.1% formic acid in acetonitrile (B) or 5 mM ammonium formate in water (C) and acetonitrile (D) at a flow rate of 0.25 mL/min. A minimum injection of 2 μL of each sample was performed after equilibration. An increasing linear gradient of solvent B (*v*/*v*) was used as follows: 0–1 min, 2% B/D; 1–9 min, 2–50% B/D; 9–12 min, 50–98% B/D; 12–13.5 min, 98% B/D; 13.5–14 min, 98–2% B/D; and 14–20 min, 2% B-positive model (14–17 min, 2% D-negative model). The ESI-MSn experiments were performed on the Thermo Q Exactive Focus mass spectrometer with a spray voltage of 3.8 kV and −2.5 kV in positive and negative modes, respectively. Sheath gas and auxiliary gas were set at 30 and 10 arbitrary units, respectively. The capillary temperature was 325 °C. The analyzer scanned a mass range of 81–1000 *m*/*z* with a mass resolution of 70,000. MS/MS data-dependent acquisition (DDA) experiments were performed using a high-energy collisional dissociation (HCD) scan. The normalized collision energy was 30 eV. Dynamic exclusion was implemented to remove some unnecessary information in MS/MS spectra. The raw data were first converted to mzXML format using MSConvert in the ProteoWizard software package (v3.0.8789) and processed using R XCMS (v3.12.0) for feature detection, retention time correction, and alignment. The settings of the most important parameters were set as follows: ppm = 15, peak width = c (5, 30), mzdiff = 0.01, method = centWave. The batch effect was then eliminated by correcting the data based on QC samples. Metabolites with RSD > 30% in QC samples were filtered and then used for subsequent data analysis. The metabolites were identified using accuracy mass and MS/MS data which were matched with HMDB (http://www.hmdb.ca, accessed on 20 May 2022), Mass Bank (http://www.massbank.jp/, accessed on 20 May 2022), KEGG (https://www.genome.jp/kegg/, accessed on 22 May 2022), Lipid Maps (http://www.lipidmaps.org, accessed on 22 May 2022), and Mzcloud (https://www.mzcloud.org, accessed on 25 May 2022), and the metabolite database was built by Panomix Biomedical Tech Co., Ltd. (Suzhou, China). The molecular weight of the metabolites was determined from the *m*/*z* (mass-to-charge ratio) of the parent ions in MS data. The molecular formula was predicted from parts per million (ppm) values and adduction and then matched to the database to enable MS identification of metabolites. At the same time, the MS/MS data from the quantitative table of MS/MS data were matched with the fragment ions and other information about each metabolite in the database to enable MS/MS identification of the metabolites. Two different multivariate statistical analysis models, unsupervised and supervised, were applied to distinguish the groups (PCA; PLS-DA; OPLS-DA) through the package R ropls (v1.22.0). The statistical significance of the *p* value was determined by a between-group statistical test. Finally, the *p* value, VIP (variable projection significance OPLS-DA), and FC (multiple of difference between groups) were combined to screen biomarker metabolites. By default, through a comparison with each control (control mouse or control samples mentioned above) when *p* value < 0.05 and VIP value > 1, the metabolites were considered to be significantly differentially expressed in *N. brasiliensis*-infected mouse or *N. brasiliensis*-derived ES products. In addition, a comparative analysis between metabolites from *N. brasiliensis*-infected mice and *N. brasiliensis*-derived ES products was performed, and the overlapping metabolites were selected for further pursuit, which could be considered as *N. brasiliensis*-derived potentially active molecules (metabolites).

### 2.4. Anti-Inflammatory Responses Detection In Vitro and In Vivo

To evaluate the anti-inflammatory effect of ES or metabolites (uridine, Sigma-Aldrich, St. Louis, MO, USA), a lipopolysaccharide (LPS, Sigma-Aldrich)-induced acute inflammation model and a bone marrow-derived macrophage model (BMDM) were used in the present study.

For the LPS model, 16 male 6-week-old C57BL/6 mice were randomly divided into four groups, including the vehicle control group (Vehicle), the LPS group (LPS), the LPS + ES group (LPS + ES), and the LPS + uridine group (LPS + UD), with 4 mice per group. LPS was used to induce acute inflammation at 2.5 mg/kg body weight using intraperitoneal injection in mice. ES or uridine were administered intragastrically 1 h before LPS injection, and 24 h and 48 h after LPS injection at 30 mg/kg (LPS + ES) and 10 mg/kg (LPS + UD) body weight, respectively, to intervene. Sterile saline was injected into the vehicle group via the same route at equal volume. Six hours after the last procedure, all mice were euthanized using carbon dioxide asphyxiation and the serum samples were collected for determination of cytokine levels using the TNF-α/IL-1β/IL-6 ELISA kit (Absin, Shanghai, China) according to the kit’s instructions (intra- and inter-assay CVs were less than 10%). In addition, the spleen samples of all mice were collected and stored at −80 °C to determine the mRNA expression levels of cytokines (IL-6, TNF-α, TGF-β, and IL-10) and inflammatory pathway-related proteins (NF- κB and pNF-κB). Reverse transcription-PCR was used to detect spleen cytokine mRNA expression levels using LightCycler 480 (Roche, Basel, Switzerland). The primers are listed in Table 1. HiScript Ⅲ RT SuperMix for qPCR (Vazyme, Nanjing, China) and ChamQ Universal SYBR qPCR Master Mix (Vazyme, Nanjing, China) were used as reagents. RNA concentrations were measured using NanoDrop2000 (Gibco-Thermo Fisher, Waltham, MA, USA) and the reaction conditions of qPCR were pre-denaturation (95 °C for 300 s) and 40 denaturation cycles (95 °C for 10 s, 60 °C for 30 s) and dissolution curve (95 °C 15 s, 60 °C 60 s, 95 °C 15 s). The protein expression of NF-κB and pNF-κB were detected using the classical Western blot analysis.

For the BMDM model, bone marrow cells were obtained from the femur and tibia of mice (C57BL/6, 6 weeks old, male) according to the standard protocol [25]. They were then stimulated with macrophage colony-stimulating factor (M-CSF, 50 ng/ml; Sigma-Aldrich, USA) over 7 days, and the bone marrow cells developed into bone marrow-derived macrophages (M0 phase). Then, different doses of uridine (20 μmol/L, 50 μmol/L, 100 μmol/L) with or without IL-4 (20 ng/mL, Abcam) were added to the BMDM culture system for the next 24 h, and the stimulated BMDM cells were collected to characterize macrophage activation (macrophage polarization). Markers of macrophage activation, including iNOS, ARG1, and CD206 mRNA expression levels, were also detected using reverse transcription-PCR, with the primers listed in Table 1.

### 2.5. Anti-Atherosclerosis Effects Evaluation in ApoE^−/−^ Mouse Model

To evaluate the anti-atherosclerosis effect of the *N. brasiliensis*-derived products (ES and uridine), the apolipoprotein E-deficient (ApoE^−/−^) mice were studied for atherosclerosis modeling (AS) by feeding the mice a high-fat diet (HFD, 21% milk fat and 0.15% cholesterol), purchased from Xietong Shengwu (Nanjing, China). Sixty male ApoE^−/−^ mice (5 weeks old) were provided by the animal center of JIPD and randomly divided into six groups of ten mice each, which were fed HFD for 12 weeks. As shown in schematic diagram of animal experiment, during AS modeling, different interventions of each group were performed twice a week, including ES interventions (30 mg/kg) via intraperitoneal (ip) or intragastric (ig) injection routes (HFD + ES ip or HFD + ES ig), uridine intervention (10 mg/kg) via the same routes (HFD + UD ip or HFD + UD ig), and intervention with 0.9% NaCl solution with the same volume as the control (HFD). In addition, ten male mice aged 5 weeks were fed normal diet (ND) for 12 weeks and used as a blank control. The body weight of all mice was monitored every week, and all mice were euthanized at the end of week 12 for evaluation of anti-atherosclerosis effects, including AS plaque areas throughout the entire artery and aortic arch, liver lipid content, associated gene expression levels, and serum lipids and cytokine levels.

For AS plaque area analysis, the entire artery was separated, stained with Oil red according to the kit protocol (Solarbio, Beijing, China), and photographed under a dissecting microscope (Olympus, Tokyo, Japan). The photos were processed using ImageJ software (V1.8.0.112). In addition, pathological sections were prepared for the aortic arch of each mouse, and HE staining was performed using the HE staining kit (Solarbio, Beijing, China). The AS plaque was observed under the microscope (Olympus, Japan). The AS plaque rate was calculated by dividing the number of AS plaques in the aorta by the total aortic area in the field of view, and three samples were collected from each group. ImageJ software was also used to calculate the plaque area in the inner wall of each aortic arch sample.

For liver lipid content analysis, liver tissue was collected from the same part of each mouse and used for the preparation of pathological sections. Additionally, HE staining was performed using the same kit and protocol as above. The lipid droplet was observed under the microscope (Olympus, Japan). The percentage of lipid droplet cells in each field of view was calculated using ImageJ software.

For lipid and cytokine level detection, each mouse underwent tail vein blood collection at the end of the week 12. The serum lipid levels, including total cholesterol (TC), low-density lipoprotein (LDL) and high-density lipoprotein (HDL), were carried out using commercial kits (Solarbio, Beijing, China) according to the kits’ protocols. The detection of serum cytokine levels, including IL-10 and TNF-α, was also performed using commercial kits (Elabsciences, Wuhan, China) according to the kits’ protocols.

For macrophage polarization and inflammation-related gene expression detection, the aortic arch of 3–4 mice from each group was used. The mRNA expression levels, including TNF-α, IL-6, ARG1, iNOS, and IL-10 genes, were detected with reverse transcription-PCR using LightCycler 480 (Roche) as mentioned above. The primers are also listed in Table 1.

### 2.6. Mitochondrial ATP-Sensitive Potassium Channel Blockage In Vitro and In Vivo

Previous studies showed that the mitochondrial ATP-sensitive potassium channel (mitoK_ATP_) was important for the anti-inflammatory effects of uridine and could be blocked with 5-hydroxydecanoate sodium (5-HD) [26,27]. To verify the role of the mitoK_ATP_ channel for uridine anti-inflammatory and anti-atherosclerosis effects, two separated experiments were carried out in the present study through channel blockage with 5-HD in vitro and in vivo.

For channel blocking in vitro, the BMDM model was used and the protocols of cell preparation, cell culture, and stimulation were the same as the protocols described above. When the bone marrow cells developed into bone marrow-derived macrophages (M0 phase), the uridine (50 μmol/L) or IL-4 (50 ng/mL) was co-cultured with BMDM using 5-HD (Sigma-Aldrich, USA), blocked or not blocked. Following the next 24 h stimulation, the cultured cells and cell culture supernatant were collected and used for gene mRNA expression and cytokine level detection, respectively. M2-macrophage-related gene markers, including YM1, CD206, ARG1, and IL-10, were detected through RT-qPCR as mentioned above. Cytokines in the supernatant, including IL-1β and IL-10, were detected using commercial kits (Elabsciences, Wuhan, China) according to the kits’ protocols.

For channel blocking in vivo, 50 male ApoE^−/−^ mice aged 5 weeks were used and divided into 5 groups of 10 mice each, including the control group with normal diet (ND), the control group with high-fat diet (HFD), 5-HD-blocking control group with high-fat diet (HFD + 5HD), uridine intervention group with high-fat diet (HFD + UD), and uridine intervention 5-HD-blocking group with high-fat diet (HFD + UD + 5HD). As shown in schematic diagram of animal experiment, the AS modeling was the same as above. Uridine intervention (with same content) was carried out through the intragastric injection route in HFD + UD or HFD + UD + 5HD groups twice a week. In the blockage groups (HFD + UD + 5HD group and HFD + 5HD group), 5-HD was injected through the intraperitoneal route (5 mg/kg) one hour before uridine intervention. The body weight of all mice was also monitored in each week and all mice were euthanized at the end of week 12 for anti-atherosclerosis effect evaluation, including AS plaque area, liver lipid content, related gene expression levels, serum lipid levels, and cytokine levels, using the same procedures described above.

### 2.7. Statistically Analysis

Graphpad Prism (V 8.0.2) and Image J (V1.8.0.112) were used for data analysis and statistics. The statistical analyses were performed using IBM SPSS Statistics, version 25 (IBM Corporation, Armonk, NY, USA). All experimental numerical results were expressed as mean ± standard error (mean ± SEMs). The *t* test was used for comparison between the two groups, and a one-way analysis of variance combined with the SNK method was used for multiple comparisons of single factors. *p* values were expressed as * *p* < 0.05 and ** *p* < 0.01.

## 3. Results

### 3.1. Uridine, a Key Metabolite Derived from Nippostrongylus brasiliensis through Metabolomics Analysis

To screen the anti-inflammatory components derived from *N. brasiliensis*, in the present study, two independent non-targeted metabolomics analyses were performed, including ES from in vitro cultured *N. brasiliensis* and intestinal contents from *N. brasiliensis*-infected mice using ultra-performance liquid chromatography-mass spectrometry (UHPLC-MS). As shown in Figure 1A, 191 metabolites with differential expression were identified in the ES group, including 67 metabolites with upregulated expression (fold change > 1 and *p* < 0.05) and 124 metabolites with downregulated expression (fold change < 1 and *p* < 0.05) compared to the control group (detailed in Appendix A). Meanwhile, compared with the mouse intestinal contents in the control group, 50 metabolites with differential expression were identified in the *N. brasiliensis*-infected group (Figure 1B), including 24 metabolites with upregulated expression (fold change > 1 and *p* < 0.05) and 26 metabolites with downregulated expression (fold change < 1 and *p* < 0.05). A further analysis was conducted for mining active metabolites derived from *N. brasiliensis* between the different metabolites from the ES group and the *N. brasiliensis*-infected group. A total of 32 metabolites overlapped in the ES group and the *N. brasiliensis*-infected group, of which 12 metabolites showed upregulated expression with fold change >1 (detailed in the Appendix A), including 1-hydroxy-2-naphthoate, uridine, kojic acid, homovanillic acid, 4,5-dihydroorotic acid, pyrrole-2-carboxylic acid, 2-keto-6-aminocaproate, gamma-aminobutyric acid, benzoate, diisobutyl phthalate, 2,3-butanediol, and pelargonic acid. Through literature tracking, the overlapping metabolites with upregulated expression showed anti-inflammatory activities, including uridine, kojic acid, gamma-aminobutyric acid, and pelargonic acid [28,29,30,31]. Considering the relative expression levels of the above metabolites in both groups, uridine showed significantly higher expression levels than others, with a fold change of 467,980 in the *N. brasiliensis*-infected group and 56,128 in the ES group (shown in Figure 1C,D), which could be selected as a key metabolite from *N. brasiliensis* and evaluated in a follow-up study.

### 3.2. Uridine Showed Anti-Inflammatory Effects In Vivo and Could Induce an M2 Macrophage Polarization Phenotype In Vitro

Uridine, a key metabolite derived from *N. brasiliensis* studied above, has been previously reported to have anti-inflammatory activities [28]. To further confirm its anti-inflammatory activities, an LPS-induced acute inflammation model and a bone marrow-derived macrophage model (BMDM) were used for further evaluation in the present study.

As shown in Figure 2, following the intraperitoneal injection of LPS, serum TNF-α and IL-6 levels were significantly increased in the LPS group compared to the vehicle group (Figure 2A,C, *p* < 0.01), while IL-1β levels had no significant change (Figure 2B, *p* > 0.05), accompanied by higher TNF-α (Figure 2D, *p* < 0.01)/IL-6 (Figure 2E, *p* < 0.01)/TGF-β (Figure 2F, *p* < 0.05) mRNA expression levels, lower mRNA expression levels of IL-10 (Figure 2G, *p* < 0.05) and higher pNF-κB (Figure 2H,I, *p* < 0.05) expression levels in the spleen compared to the vehicle group, indicating a successful induction of acute inflammation in vivo. However, there was a significant decrease in serum TNF-α (Figure 2A, *p* < 0.01 and <0.05, respectively)/IL-1β (Figure 2B, *p* < 0.05)/IL-6 (Figure 2C, *p* < 0.05 or <0.01) pNF-κB (Figure 2H,I, *p* < 0.05 and < 0.01,respectively) and TNF-α (Figure 2D, *p* all < 0.01)/IL-6 (Figure 2E, *p* all < 0.01) mRNA expression levels in the LPS + ES or LPS + UD group compared to those in the LPS group. Meanwhile, there was a significant increase in the mRNA expression levels of TGF-β (Figure 2F, *p* all < 0.05) and IL-10 (Figure 2G, *p* < 0.01 and <0.05, respectively) in the LPS + ES or LPS + UD group, compared with the LPS group. No significant differences were observed between the LPS + ES and LPS + UD groups in serum cytokine levels (TNF-α/IL-1β/IL-6) and cytokine mRNA expression levels (TNF-α/IL-6/TGF-β/IL-10) and pNF-κB expression levels (Figure 2A–I). There was also no significant difference in NK-κB expression levels in all groups. The above results indicated that *N. brasiliensis*-derived ES products or uridine could induce anti-inflammatory activity in vivo, manifested by reduced expression levels of pro-inflammatory cytokines or inflammatory pathway proteins and increased expression levels of anti-inflammatory cytokines.

Macrophages play an important role in inflammation regulation through various phenotypes, including classical activated macrophages (M1) with pro-inflammatory functions and alternatively activated macrophages (M2) with anti-inflammatory functions [32,33]. To further evaluate the activities of uridine on macrophage differentiation in the present study, the BMDM model was used in vitro and the macrophage phenotype was detected based on the mRNA expression levels of the M2 surface markers (ARG1 and CD206). As shown in Figure 2J, uridine at a concentration of 100 μmol/L could induce an M2 phenotype with significantly higher mRNA expression levels of ARG1 (*p* < 0.05) compared to the control (0 μmol/L). Furthermore, at a concentration of 100 μmol/L, a synergistic effect with IL-4 on the M2 phenotype induction of uridine was observed, which was manifested in significantly higher mRNA expression levels of ARG1 (Figure 2K, *p* < 0.01). However, there were no significant differences in CD206 mRNA expression levels in all groups. The above results suggest that uridine could induce an anti-inflammatory macrophage phenotype (M2 polarization) and show a synergistic effect for M2 polarization with IL-4 in vitro.

### 3.3. Uridine Showed Anti-Atherosclerosis Effects in ApoE^−/−^ Mouse Model through Inducing Anti-Inflammatory Responses

The above results confirmed the anti-inflammatory effect of uridine in vitro and in vivo. To further evaluate the anti-atherosclerosis effect, ApoE^−/−^ mice which were fed a high-fat diet were used, and ES/uridine intervention was performed using intraperitoneal (ip) or intragastric (ig) administration (Figure 3A); the resulting changes including AS lesion/plaque area, liver lipid content, serum lipid/cytokine levels, and associated gene expression levels were observed and calculated for the evaluation of the anti-atherosclerosis effect. As shown in Figure 3, ApoE^−/−^ mice (HFD group), which were fed a high-fat diet, showed significant weight gain within 12 weeks (Figure 3B,F), and produced significantly more AS lesion areas in the entire artery (Figure 3D,H, *p* all < 0.01), larger AS plaque areas in the aortic arch (Figure 3E,I, *p* all < 0.01), and higher fatty areas in the liver (Figure 3C,G, *p* all < 0.01) in comparison to the group fed with a normal diet (ND group), which indicated the successful modeling of atherosclerosis in ApoE^−/−^ mice. Compared to the HFD group, there were significant differences in the ES and uridine intervention groups (HFD + ES ip, HFD + UD ip, HFD + ES ig, and HFD + UD ig), including lower body weight (Figure 3B,F; *p* < 0.05, <0.01, <0.05, and <0.05, respectively), less AS lesion areas in the entire artery (Figure 3D,H, *p* all <0.05), smaller AS plaque areas in the aortic arch (Figure 3I, *p* all < 0.01), and lower fatty areas in the liver (Figure 3C,G, *p* all < 0.01).

Serum lipid levels are shown in Figure 4A–C. Compared to the ND group, there was a significant difference in the HFD group, including higher levels of total cholesterol (Figure 4A, *p* < 0.01) and low-density lipoprotein (Figure 4B, *p* < 0.01), and lower levels of high-density lipoprotein (Figure 4C, *p* < 0.01). Compared to the HFD group, there were significant differences in the ES and uridine intervention groups (HFD + ES ip, HFD + UD ip, HFD + ES ig, and HFD + UD ig), including lower total cholesterol levels (Figure 4A, *p* < 0.01, <0.01, <0.01, and <0.05, respectively), lower low-density lipoprotein levels (Figure 4B, *p* all <0.05), and higher high-density lipoprotein levels (Figure 4C, *p* all < 0.05). However, no significant differences were observed between the HFD group and the HFD + ES ig or HFD + UD ig group in low-density lipoprotein levels and high-density lipoprotein levels, respectively (Figure 4B,C).

Serum cytokine levels are shown in Figure 4D,E. Compared with the ND group, serum TNF-α levels were significantly increased (Figure 4D, *p* < 0.01), while IL-10 levels were significantly decreased in the HFD group (Figure 4E, *p* < 0.05). Compared with the HFD group, serum TNF-α levels in the ES and uridine intervention groups were all significantly decreased (Figure 4D, *p* < 0.01, <0.05, <0.05 and <0.05, respectively), while serum IL-10 levels were all significantly increased in ES and uridine intervention groups (Figure 4E, *p* all <0.05).

Figure 4F–H showed macrophage polarization and inflammation-related gene expression levels in the aortic arch. Compared with the ND group, the TNF-α and IL-6 mRNA expression levels were significantly increased in the HFD group (Figure 4F,G, *p* all < 0.01), while the ARG1 mRNA expression level was significantly reduced in the HFD group (Figure 4H, *p* < 0.05). However, there were no significant differences in iNOS and IL-10 mRNA expression levels between the ND group and the HFD group. Compared with the HFD group, TNF-α and IL-6 mRNA expression levels were decreased significantly in the ES and uridine intervention groups (Figure 4F, *p* < 0.05, <0.01, <0.05 and <0.01, respectively; Figure 4G, *p* < 0.05, <0.01, <0.01 and <0.05, respectively), while ARG1 mRNA expression levels were increased significantly in the ES and uridine intervention groups (Figure 4H, *p* < 0.05, >0.05, >0.05 and <0.01, respectively). However, there were no significant differences in iNOS and IL-10 mRNA expression levels between the HFD group and ES/uridine intervention groups.

The above results indicated that *N. brasiliensis* derived ES products and uridine exhibited anti-atherosclerosis effects via intraperitoneal or intragastric administration routes in mouse model, which could decrease the liver lipid content and alleviate the formation of atherosclerosis through inducing expression of anti-inflammatory agents and inhibition of pro-inflammatory responses.

### 3.4. Anti-Inflammatory Responses and Anti-Atherosclerosis Effects Generated by Uridine Were Partially Dependent on Mitochondrial ATP-Sensitive Potassium Channel

As a small biological molecule, uridine could transform to uridine diphosphate (UDP) following enter into cell through simple diffusion, which plays biological role through activation of the mitochondrial ATP-dependent potassium channel (mitoK_ATP_) [34]. Furthermore, activation of mitoK_ATP_ could induce anti-inflammatory responses by inhibiting NF-κB transcription and activation of M2 macrophages, which could be blocked by 5-hydroxydecanoate (5-HD) [35,36]. To inquiry the mechanism of anti-inflammatory and anti-atherosclerosis effects by uridine, channel blockage was carried out by using 5-HD in BMDM and ApoE^−/−^ mouse models for effect evaluation.

As shown in Figure 5, uridine or IL-4 could induce M2 polarization similar to the above in the BMDM model, with significantly increased Ym1, CD206, ARG1 mRNA expression levels and IL-10 levels. Furthermore, the levels of ARG1, Ym1, CD206 mRNA expression and IL-10 were significantly lower after blocking with 5-HD in uridine stimulation BMDM modules (*p* all < 0.05), when compared with that in no 5-HD blockage. However, there were no significant differences of Ym1, CD206, ARG1 mRNA expression and IL-10 levels in IL-4 stimulation modules with 5-HD blockage or not. Also, no significant differences in IL-10 mRNA expression and IL-1β levels were observed in IL-4 or uridine stimulation modules with or without 5-HD blockade. The results indicated that M2 polarization induction by uridine depends on the activation of mitoK_ATP_ channel in BMDM models.

In ApoE^−/−^ mouse model, when compared to the ND group, similar results were observed in the HFD group, with significantly increased body weight (Figure 6B,F, *p* < 0.05), AS lesion areas throughout the entire artery (Figure 6D,H, *p* < 0.05), AS plaque areas in the aortic arch (Figure 6E,I, *p* < 0.01), areas of higher fat content in the liver (Figure 6C,G, *p* < 0.01), total cholesterol levels (Figure 7A, *p* < 0.05), low-density lipoprotein levels (Figure 7B, *p* < 0.05), TNF-α levels (Figure 7D, *p* < 0.05), TNF-α, IL-6 and IL-10 mRNA expression levels (Figure 7F–H, *p* < 0.01, <0.01 and <0.05, respectively); meanwhile with significantly decreased high-density lipoprotein levels (Figure 7C, *p* < 0.05), IL-10 levels (Figure 7E, *p* < 0.05), ARG1 mRNA expression levels (Figure 7I, *p* < 0.05), which also indicated the successful AS modeling.

Compared to the HFD group, similar results were also observed in the HFD + UD group, with decreased body weight (Figure 6B,F, *p* < 0.01), AS lesion areas throughout the artery (Figure 6D,H, *p* < 0.05), AS plaque areas in the aortic arch (Figure 6E,I, *p* < 0.05), fatty areas in the liver (Figure 6C,G, *p* < 0.05), total cholesterol levels (Figure 7A, *p* < 0.05), low-density lipoprotein levels (Figure 7B, *p* < 0.01), TNF-α levels (Figure 7D, *p* < 0.05), and TNF-α and IL-6 mRNA expression levels (Figure 7F,G, *p* < 0.05 and <0.01, respectively); significantly increased high-density lipoprotein levels (Figure 7C, *p* < 0.05), IL-10 levels (Figure 7E, *p* < 0.05), and IL-10 and ARG1 mRNA expression levels (Figure 7H,I, *p* < 0.01 and <0.05) were observed, which also indicated the anti-atherosclerosis effect of uridine in ApoE^−/−^ mice.

Compared with the HFD + UD group, the 5-HD-blocked group (HFD + UD + 5HD) showed significantly higher AS lesion areas in the entire artery (Figure 6D,H, *p* < 0.05), AS plaque areas in the aortic arch (Figure 6E,I, *p* < 0.05), fat content in the liver (Figure 6C,G, *p* < 0.05), serum low-density lipoprotein levels (Figure 7B, *p* < 0.05), TNF-α levels (Figure 7D, *p* < 0.05), and TNF-α and IL-6 mRNA expression levels (Figure 7F,G, *p* < 0.05 and <0.01, respectively); significantly lower serum high-density lipoprotein levels (Figure 7C, *p* < 0.05), IL-10 levels (Figure 7E, *p* < 0.05), and IL-10 and ARG1 mRNA expression levels (Figure 7H,I, *p* all < 0.05) were observed. However, no significant differences were observed in body weight and serum total cholesterol levels between the HFD + UD group and the HFD + UD + 5HD group. The above results suggested that the blockage of mitoK_ATP_ channels could impair the anti-inflammatory and anti-atherosclerosis effects of uridine in ApoE^−/−^ mice.

## 4. Discussion

Based on the “hygiene hypothesis”, helminth infections or derived products showed significant effects on inflammation regulation and symptom relief in various models of inflammatory-related disease [10,37]. And the screening and identification of helminth-derived active molecules from helminths have been the main topics at present [38,39]. In addition to those before the identified helminth-derived peptides or proteins with anti-inflammatory effects [21,38], our research focused on the helminth-derived small molecular metabolites due to their low antigenicity and good druggability. Our results showed that uridine, as a highly expressed metabolite from *N. brasiliensis*, exhibited ideal anti-inflammatory effects and anti-atherosclerosis effects in in vitro and in vivo models, with its biological effects dependent on the activation of the mitoK_ATP_ channel. Our present study may provide a novel strategy for mining active molecules from helminths and ultimately investigate a new molecule (uridine) with anti-inflammatory and anti-atherosclerotic effects.

A previous study reported the metabolomics profiling of the excretory–secretory (ES) products of helminths (*Nippostrongylus brasiliensis* and *Trichuris muris*) and up to 17 metabolites known to have various pharmacological activities, including wound-healing, anti-inflammatory, and cardioprotective activities [22]. Our results also revealed partially overlapping metabolites with different activities from ES products reported in previous studies including uridine [22,24], which further indicates the important key position of ES products for helminth-derived active molecules or the mining of pharmaceutical molecules. In addition to the metabolic analysis for *N. brasiliensis* ES products, to further investigate the *N. brasiliensis*-derived metabolites with activities, another metabolic analysis was performed for the intestinal contents of *N. brasiliensis*-infected mice, and the overlapping metabolites were selected for follow-up. The comparative analysis could be more accurate for screening helminth-derived molecules. However, in the present study, only the untargeted metabolomics detection method was used, and the relative fold changes were used for further screening. The targeted metabolomics method will be added for further verification. Meanwhile, in the present study, only overlapping metabolites with expression levels (fold change) were used for the screening of potentially active metabolites, and the metabolic pathway analysis based on the KEGG results would provide further information for the screening of active metabolites. Through literature tracking, six overlapping metabolites with high fold changes showed anti-inflammatory activity, and only uridine was selected for further evaluation based only on its high expression levels in two metabolic analyses. The others may also have good activities and should be evaluated in the near future.

Uridine, a pyrimidine nucleoside involved in the synthesis of cellular RNA and biological membranes, is widely distributed in host blood and extracellular tissue fluid and shows good anti-inflammatory and symptom-ameliorating effects in asthma, Sephadex gel-induced pneumonia, arthritis, and inflammatory bowel disease models [28,40,41,42]. The present study also demonstrated the proper anti-inflammatory effect of uridine in a LPS-induced acute inflammation model. Furthermore, it could stimulate an anti-inflammatory phenotype of macrophages (M2 polarization) in vitro, which may be more important for AS interventions. However, the effect of AS intervention on uridine has not been reported. In ApoE^−/−^ mouse models, two routes of administration of ES or uridine were used for anti-atherosclerosis evaluation and significant protective effects were produced by ES and uridine, which were manifested in the induction of an anti-inflammatory environment and the alleviation of AS development, which is similar to our previous study with *N. brasiliensis*-infected interventions [20]. No differences were observed between different routes of administration. It should be noted that uridine has been reported to have the function of regulating lipid metabolism, which partially explains why it has certain atheroprotective effects [40]. However, treatment effects or interventions against the AS development of ES or uridine need to be further investigated in the future. It had been reported that macrophages play an important role in the occurrence and development of AS, especially with alternative activated macrophages (M2 polarization) [43,44]. In our present study, only the macrophage activation markers were detected in the mouse aortic arch, including ARG1 and CD206 mRNA expression levels, and observed the increasing expression of M2 macrophages. The addition of the in situ detection of macrophage markers would be more suitable for deciphering the anti-atherosclerosis effect in the mouse model.

As a kind of small molecule, uridine could enter the cell through simple diffusion and convert into uridine diphosphate, which exerts biological effects through the activation of the mitochondrial ATP-dependent potassium channel (mitoK_ATP_). Previous studies had also reported that the activation of the mitoK_ATP_ channel could induce the activation of M2 macrophages for anti-inflammatory effects by inhibiting the transcription of NF-κB signals, which could be blocked by 5-hydroxydecanoate (5-HD) [35,36]. Our present results also indicated that the induction of M2 macrophages with uridine could also be blocked with 5-HD in vitro, which further confirmed that the induction of M2 macrophage activation depends on the mitoK_ATP_ channel activation. Meanwhile, in the ApoE^−/−^ mouse model, the anti-atherosclerosis effect of uridine could also be blocked with 5-HD, leading to a corresponding reduction in anti-inflammatory agents, indicating that the anti-atherosclerosis effect of uridine also depends on the mitoK_ATP_ channel activation. It should also be noted that 5-HD, as a specific inhibitor of potassium transport in mitochondria with high selectivity, has exhibited a weakened protective effect of uridine to AS in the present study, but its detailed dose relationship still needs further verification. It has also been found that 5-HD has K_ATP_ channel-independent targets in the heart which indicated that the attenuation of the protective effect of uridine could not be entirely attributed to the role of the mitoK_ATP_ channel through 5-HD [45]. We should still continue to supplement the detection of mitochondria and exclude the effects of other channels after they are blocked with 5-HD in the future. Furthermore, the gap between the mitoK_ATP_ activation and M2 macrophage activation or anti-atherosclerosis outcomes need to be further clarified. MitoK_ATP_ channels have been reported to be widely distributed in the inner mitochondrial membrane and play a key role in mitochondrial physiology and inflammatory pathological processes [46]. Previous studies have reported that the activation of mitoK_ATP_ channels could effectively ameliorate mitochondrial dysfunction caused by temporal lobe epilepsy, and play an important role in regulating mitochondrial function [47]. During the occurrence and development of atherosclerosis, the dysfunction of cell mitochondria (especially macrophages) is an important pathophysiological process, and resolving mitochondrial dysfunction by using dimethyl methylate could effectively alleviate the occurrence of atherosclerosis [48,49]. We hypothesized that mitochondria may be the main target organelle of uridine for its anti-inflammatory and anti-atherosclerotic effects through the activation of mitoK_ATP_ channels, which needs to be further investigated and confirmed in the near future.

## 5. Conclusions

In conclusion, the present study found that uridine, a small metabolite derived from *Nippstrongylus brasiliensis*, exhibited potential mitoK_ATP_ channel-dependent anti-inflammatory and anti-atherosclerotic effects. These findings may pave the way for the druggability of metabolites derived from helminths and open up new avenues for atherosclerotic intervention.

## Figures and Tables

**Figure 1 biomolecules-14-00672-f001:**
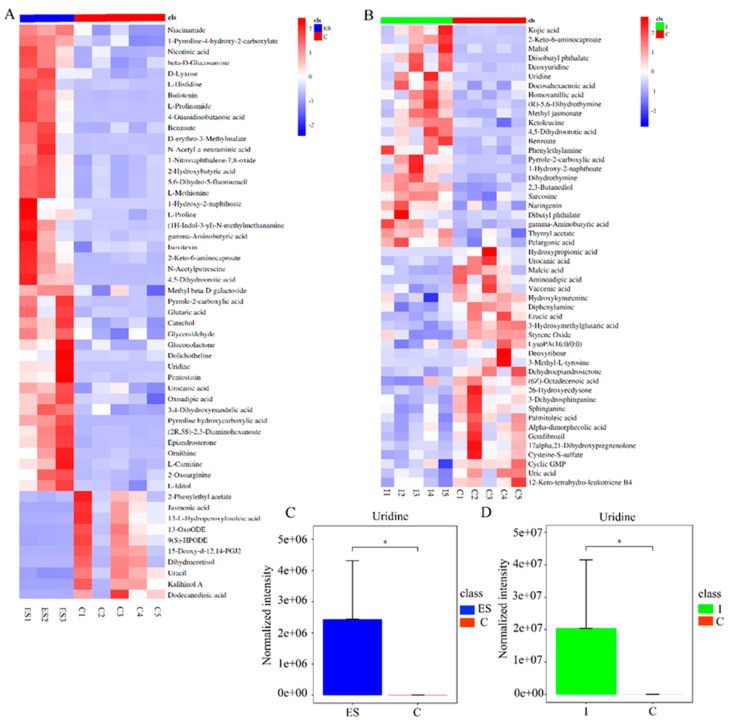
Screening of anti-inflammatory components derived from Nb through UHPLC-MS. (**A**) Heatmaps of different metabolites between the ES-cultured and control group. (**B**) Heatmaps of different metabolites treated by intestinal contents of mice infected with Nb. (**C**,**D**) Relative expression levels of uridine including ES and Nb infected groups compared with controls. * *p* < 0.05.

**Figure 2 biomolecules-14-00672-f002:**
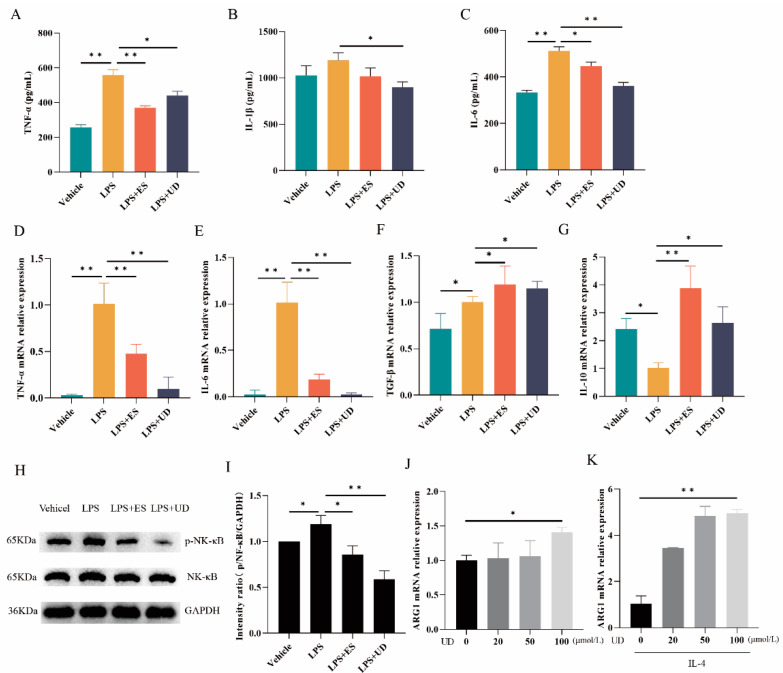
Anti-inflammatory activities observation for the screened uridine via LPS model and BMDM model. (**A**–**C**) Serum levels of TNF-α/IL-1β/IL-6 in LPS model obtained using ELISA analysis, respectively (*n* ≥ 6); (**D**–**G**) Serum mRNA relative expressions of TNF-α/IL-6/TGF-β/IL-10 in LPS model obtained using RT-qPCR, respectively (*n* ≥ 6). (**H**,**I**) Western blot analysis for phosphorylation of NF-κB in LPS model and statistical histogram of gel electrophoresis bands (*n* ≥ 6). Original images can be found in Appendix A. (**J**,**K**) Serum mRNA relative expressions of ARG1 in BMDM model obtained using RT-qPCR (*n* ≥ 6). * *p* < 0.05 and ** *p* < 0.01.

**Figure 3 biomolecules-14-00672-f003:**
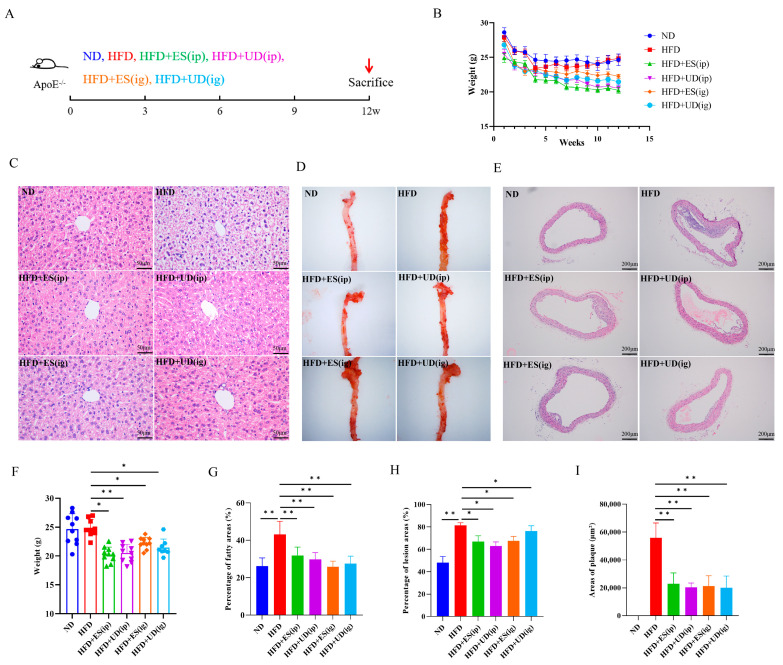
Anti-atherosclerosis effects observation following different interventions in AS mouse model. (**A**) Schematic diagram of experimental design of ES and uridine administration in AS model. (**B**,**F**) Body weight changes during the experimental period and comparison on 12th week between groups (*n* ≥ 6). (**C**,**G**) HE staining of mouse liver under 400× magnification and statistics on percentage of fatty areas between groups (*n* ≥ 5). (**D**,**H**) Oil red staining of whole arterial and statistics of lesion areas between groups (*n* ≥ 5). (**E**,**I**) HE staining of plaques within the aortic arch under 100× magnification and statistics of plaque area between groups (*n* ≥ 5). * *p* < 0.05 and ** *p* < 0.01.

**Figure 4 biomolecules-14-00672-f004:**
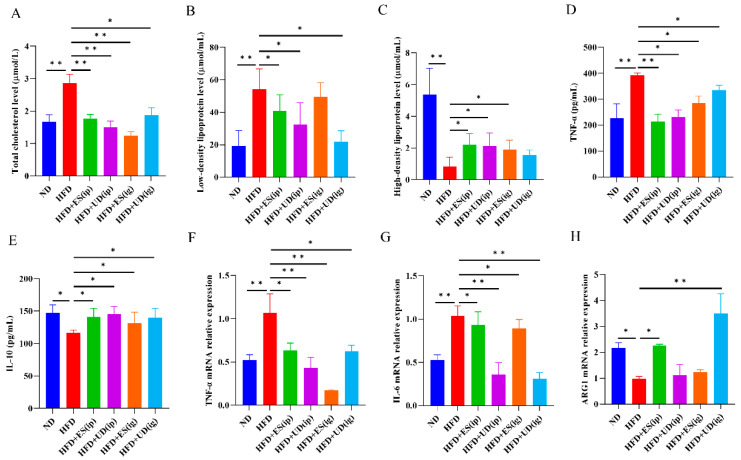
Observation of blood lipid levels and inflammatory related factor expression levels in AS mouse model following different interventions. (**A**–**C**) Serum levels of total cholesterol/LDL/HDL (*n* ≥ 6). (**D**,**E**) Serum levels of TNF-α and IL-10 in AS model by ELISA (*n* ≥ 6). (**F**–**H**) TNF-α/IL-6/ARG1 mRNA relative expressions levels in aortic arch by RT-qPCR (*n* ≥ 6). * *p* < 0.05 and ** *p* < 0.01.

**Figure 5 biomolecules-14-00672-f005:**
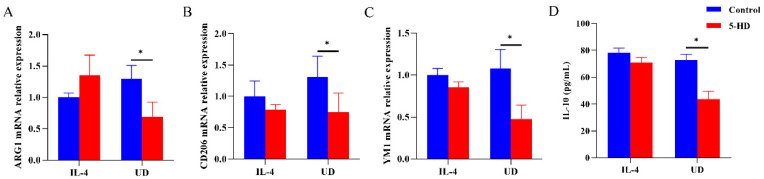
Role of mitoKATP channel for the uridine’s anti-inflammatory effects in BMDM model through blocked by 5-HD. (**A**–**C**) ARG1/CD206/YM1 mRNA relative expressions levels in BMDM model blocked by 5-HD through RT-qPCR (*n* ≥ 6). (**D**) Serum levels of IL-10 in BMDM model blocked by 5-HD through ELISA (*n* ≥ 6). * *p* < 0.05.

**Figure 6 biomolecules-14-00672-f006:**
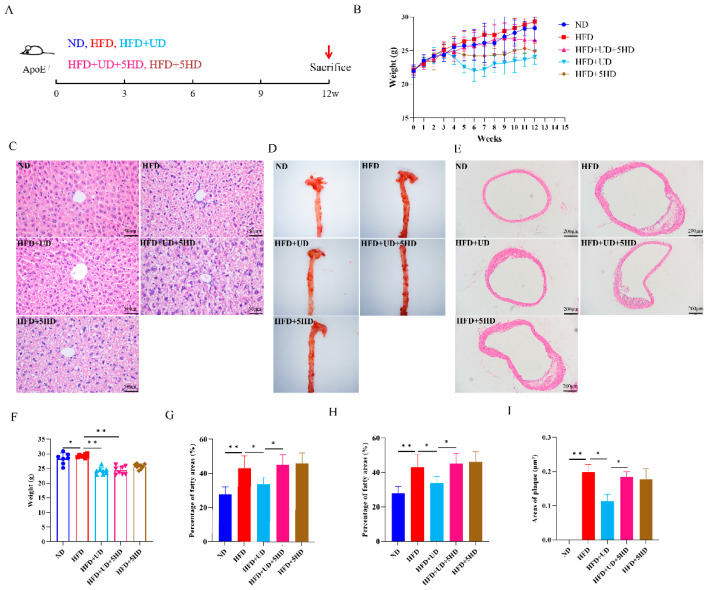
Anti-atherosclerosis effects observation for uridine in AS mouse model following channel blockage by 5-HD. (**A**) Schematic diagram of experimental design of uridine and 5-HD blockage in AS model. (**B**,**F**) Body weight changes during the experimental period and comparison on 12th week between groups (*n* ≥ 6). (**C**,**G**) HE staining of mouse liver under 400× magnification and statistics on percentage of fatty areas between groups (*n* ≥ 5). (**D**,**H**) Oil red staining of whole arterial and statistics of lesion areas between groups (*n* ≥ 5). (**E**,**I**) HE staining of plaques within the aortic arch under 100× magnification and statistics of plaque area between groups (*n* ≥ 5). * *p* < 0.05 and ** *p* < 0.01.

**Figure 7 biomolecules-14-00672-f007:**
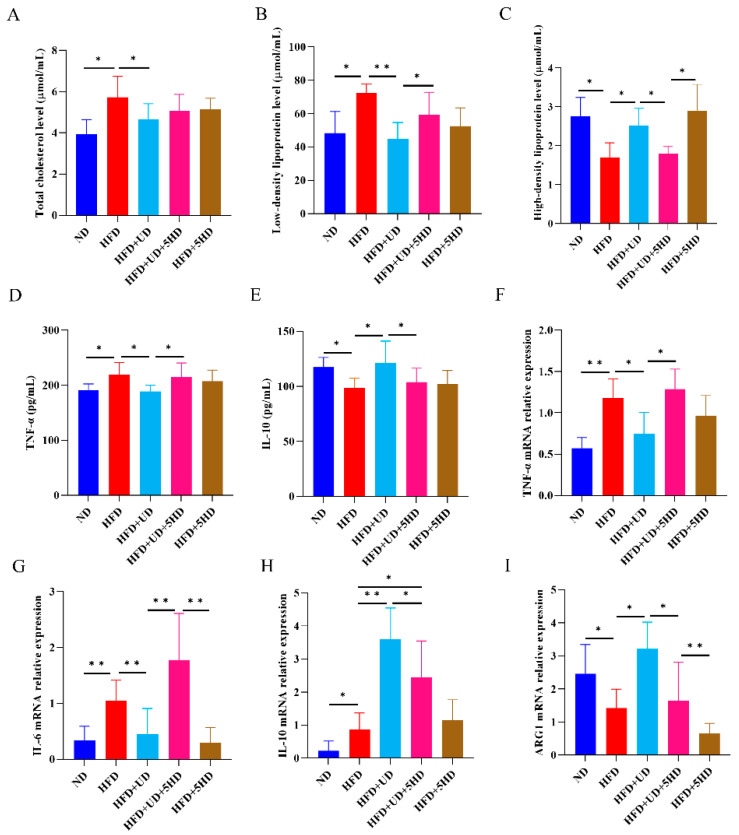
Observation of blood lipid levels and inflammatory-related factor expression levels in AS mouse model following channel blockage with 5-HD. (**A**–**C**) Serum levels of total cholesterol/LDL/HDL (*n* ≥ 6). (**D**,**E**) Serum levels of TNF-α and IL-10 in AS model obtained using ELISA (*n* ≥ 6). (**F**–**I**) TNF-α/IL-6/IL-10/ARG1 mRNA relative expressions levels in aortic arch obtained using RT-qPCR (*n* ≥ 6). * *p* < 0.05 and ** *p* < 0.01.

**Table 1 biomolecules-14-00672-t001:** Primers used in present study.

Primers	Forward Primer (5′-3′)	Reverse Primer (5′-3′)
*IL-1β*	TGTCTTGGCCGAGGACTAAGG	TGGGCTGGACTGTTTCTAATGC
*IL-6*	AGGGTCTGGGCCATAGAACT	CCACCACGCTCTTCTGTCTAC
*TNF-α*	CTGAACTTCGGGGTGATCGG	GGCTTGTCACTCGAATTTTGAGA
*iNOS*	CCTTCCGAAGTTTCTGGCAGCAGC	GGCTGTCAGAGCCTCGTGGCTTTGG
*IL-10*	AGCCTTATCGGAAATGATCCAGT	GGCCTTGTAGACACCTTGGT
*TGF-β*	ATTCCTGGCGTTACCTTGG	AGCCCTGTATTCCGTCTCCT
*ARG1*	GTGAAGAACCCACGGTCTGT	GCCAGAGATGCTTCCAACTG
*YM1*	CAGGTCTGGCAATTCTTCTGAA	CTCTTGCTCATGTGTGTAAGTGA
*CD206*	TCTTTGCCTTTCCCAGTCTCC	TGACACCCAGCGGAATTTC
*GAPDH*	TGGCCTTCCGTGTTCCTAC	GAGTTGCTGTTGAAGTGGCA

## Data Availability

Data generated or analyzed during this study are included in the article and Appendix A are available from the corresponding author on reasonable request.

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
