# Peer review of "Anti-Inflammatory Responses Produced with Nippostrongylus brasiliensis-Derived Uridine via the Mitochondrial ATP-Sensitive Potassium Channel and Its Anti-Atherosclerosis Effect in an Apolipoprotein E Gene Knockout Mouse Model"

_biomolecules, 2024, doi:10.3390/biom14060672_

Round 1
Reviewer 1 Report
Comments and Suggestions for Authors
I have thoroughly enjoyed reviewing your manuscript (ID 3021225) and want to command authors on successful study expanding our knowledge on benefits of using products of helminth infection, such as uridine, as a protective measure from Atherosclerosis. While I admire novelty and clinical relevance of the study and wish to see it published in Biomolecules, there are few minor concerns I would like authors to address. Specifically, I find it rather challenging to examine Fig. 3A and 3B due to their small sizes. I also want to advocate against selecting yellow font in Fig. 3A and light green font in Fig. 6A as the use of those fonts strain reader's eyes.
Otherwise, I would like to send my best wishes to authors.
Author Response
Reviewer1
I have thoroughly enjoyed reviewing your manuscript (ID 3021225) and want to command authors on successful study expanding our knowledge on benefits of using products of helminth infection, such as uridine, as a protective measure from Atherosclerosis. While I admire novelty and clinical relevance of the study and wish to see it published in Biomolecules, there are few minor concerns I would like authors to address. Specifically, I find it rather challenging to examine Fig. 3A and 3B due to their small sizes. I also want to advocate against selecting yellow font in Fig. 3A and light green font in Fig. 6A as the use of those fonts strain reader's eyes.
Reply: Really much thanks for your kind comments. We have made relevant improvements to optimize the visual effects of images according to your suggestions.
Reviewer 2 Report
Comments and Suggestions for Authors
In the present study authors examined the effect of hemintic (Nippostrongylus brasiliensis) secretory/excretory products on the development of atherosclerosis and related inflammation in high fat diet fed apoE knockout mice. They identified uridine as the major metabolite responsible for antiinflammatory activity and its mechanism of action included activation of mitochondrial ATP-sensitive potassium channels.
After reading this manuscript I have the following comments:
1. Groups of the animals used in the experiments should be specified in the Abstract.
2. All abbreviations used in the Abstract should be explained in the Abstract itself. The Abstract should be self-explanatory without referring to the main text.
3. Limits of quantification as well as intra- and inter-assay CVs for all ELISA assays should be specified.
4. More details about RT-qPCR such as methods used to measure RNA concentration and quality, time and temperatures of consecutive cycle phases should be provided.
5. Line 241: were wild-type mice used as the control?
6. LIne 262: the method of blood sampling for serum collection should be specified.
7. Total number of mice used in the experiments should be specified.
8. How specific is 5-HD as the mitochondrial ATP potassium channel blocker, especially at doses used in vivo in the present study?
9. What clinical implications of the findings are proposed? Would it be reasonable to use uridine or complex ES mixture?
10. Due to huge effect of uridine in the LPS model, could it be used also in acute inflammatory conditions such as sepsis?
11. There are important differences in atherosclerosis development between males and females. The present experiments were performed only in males; can the results be extrapolated to females?
12. Statistical analysis: was normality of data distribution verified?
13. In the high fat model uridine reduced body weight gain - what is the mechanism of this effect? Could the effect on body weight mediate, at least in part, the effect on atherosclerosis? It would be reasonable to include the pair-fed group to address this issue.
14. The conclusion that activation of mtK-ATP channels should be treated with caution as the effect on channel activity was not examined and only one approach was used to address their involvement. It would be reasonable to examine the effect of 5-HD on antiatherosclerotic effect of other treatment methods to confirm its specificity regarding uridine.
Author Response
Reviewer2
- Groups of the animals used in the experiments should be specified in the Abstract.
Reply: Thanks for reviewer’s comments. We have adjusted the description in the abstract.
- All abbreviations used in the Abstract should be explained in the Abstract itself. The Abstract should be self-explanatory without referring to the main text.
Reply: Thanks. We have explained the abbreviations in the Abstract.
- Limits of quantification as well as intra- and inter-assay CVs for all ELISA assays should be specified.
Reply: Thanks for your advice. We have annotated it in the article.
- More details about RT-qPCR such as methods used to measure RNA concentration and quality, time and temperatures of consecutive cycle phases should be provided.
Reply: Thanks for your comments. We have supplemented descriptions about RNA concentration determination and reaction conditions in the text.
- Line 241: were wild-type mice used as the control?
Reply: Thanks. ApoE-/- mice fed with normal diet were chosen as the control group for observation rather than wild-type mice in this study.
- Line 262: the method of blood sampling for serum collection should be specified.
Reply: Thanks for your comments and we have explained the describe of blood collection method.
- Total number of mice used in the experiments should be specified.
Reply: Thanks. We mentioned it in line 238 of the article.
- How specific is 5-HD as the mitochondrial ATP potassium channel blocker, especially at doses used in vivo in the present study?
Reply: Thanks for the reviewer’s comments. 5-HD is a specific inhibitor of potassium transport in mitochondria with high selectivity and exhibited a weakened protective effect of uridine to AS in the present study. However, its detailed dose relationship still needs further verification.
- What clinical implications of the findings are proposed? Would it be reasonable to use uridine or complex ES mixture?
Reply: Very much thanks for your suggestion. According to our study and previous research, the products derived from hookworms such as ES could exert certain anti-inflammatory effects. It has been proved that uridine as one of its components also plays an analogous pharmacological role. These findings could provide an experimental evidence for immunotherapy of hookworms and mass experiments are still needed for pharmacological and clinical validation.
- Due to huge effect of uridine in the LPS model, could it be used also in acute inflammatory conditions such as sepsis?
Reply: Really thanks for your opinion. Although our research mainly focused on chronic inflammatory diseases such as atherosclerosis, inflammatory bowel disease and melanoma, we believe that uridine can also exert certain therapeutic effects in acute diseases
- There are important differences in atherosclerosis development between males and females. The present experiments were performed only in males; can the results be extrapolated to females?
Reply: Thanks for your highly technical comments. The incidence rate of atherosclerosis in the population varies greatly between gender and the reason we chose male mice was to establish the model more quickly and stably, as well as to eliminate interference from pregnant female mice. Thus, the conclusion in this study cannot be fully extrapolated to females.
- Statistical analysis: was normality of data distribution verified?
Reply: The statistical analyses were performed using IBM SPSS Statistics 25 software. Normal distribution was a prerequisite operation for the statistical method mentioned in this study, and it has been verified through validation.
- In the high fat model uridine reduced body weight gain - what is the mechanism of this effect? Could the effect on body weight mediate, at least in part, the effect on atherosclerosis? It would be reasonable to include the pair-fed group to address this issue.
Reply: Thanks for reviewer’s comments. Weight loss may be related to the regulation of energy and lipid metabolism by uridine and the promotion of lipid metabolism can improve the formation of intravascular plaques. Body weight, as a routine indicator of animal models, was mainly used in this study to measure the differences in growth status caused by intervention conditions between groups.
- The conclusion that activation of mitoK-ATP channels should be treated with caution as the effect on channel activity was not examined and only one approach was used to address their involvement. It would be reasonable to examine the effect of 5-HD on anti-atherosclerotic effect of other treatment methods to confirm its specificity regarding uridine.
Reply: Thanks for the reviewer’s suggestion. Although it cannot be entirely attributed to the role of mitoKATP channel that 5-HD as a specific inhibitor of potassium transport in mitochondria with high selectivity has exhibited a weakened protective effect of uridine to AS in the present study. We should also continue to supplement the detection of mitochondria and exclude the effects of other channels after blocked by 5-HD in the future.
Reviewer 3 Report
Comments and Suggestions for Authors
This manuscript by Zhang et al. describes the possible anti-atherosclerotic effects of secretion/excretion products of N. brasiliensis helminth parasite and, in particular, of uridine as one of the main metabolites with anti-inflammatory activity that they contain. Additionally, it shows evidence of the possible involvement of the mitochondrial ATP-sensitive potassium ion channel as a mediator of the anti-inflammatory effects of uridine.
The experiments described within are well designed and their results provide good evidence for the anti-inflammatory and anti-atherosclerotic activity of uridine. However, I have some concerns to express that the authors should address to improve the manuscript.
As minor points that are easy to resolve, I want to draw attention to the following:
-
In the Abstract the abbreviation ES is used without a prior definition, which makes it difficult to understand what the authors are referring to.
-
In lines 222, 283, 379, 381 and in Figure 2 μmol is used to refer to a concentration. This should be expressed in the appropriate units.
The in vivo and in vitro experiments shown point toward an anti-inflammatory activity of N. brasiliensis ES products and uridine, but, in in vivo experiments using APOE KO mice on high fat diet as a model for atherosclerosis development, the authors focus their interpretation of the results (reduced atherogenesis) on the anti-inflammatory associated effects of the treatment. Nevertheless, it is also shown that uridine treated animals show a serum lipid profile modifications, as well as a lower body weight and reduced fatty liver signs. These facts should be taken into consideration to explain the reduced development of atherosclerotic lesions, as all of them can contribute to the output of the treatment.
I suggest considering the potential effect of the dose of uridine used in the experiment on overall lipid and energy metabolism to give rise, at least partially, to an atheroprotective effect. This could be included in the Discussion section and the manuscript already contains references supporting this point (reference 40).
Regarding the results of the in vivo experiments, it is surprising to me the different behaviour of the animals' body weight throughout the treatment period: in Figure 3B all groups show a reduction in body weight, although it is more pronounced in the mice treated with ES or uridine; in Figure 6B there is an increase of body weight in all groups except those treated with uridine and those treated with 5HD, whose body weight seems to stabilise below the non-treated animals from 4th week, and uridine+5HD treated mice, whose body weight rises until 9th week and then stabilises. It is specially discording among both experiments the fact that for ND and HFD the body weight decreases in one case (Figure 3B) or increases in the other (Figure 6B), assuming that the animals had the same genetic background (APOE KO), the same initial age (5 weeks), and were subjected to similar experimental conditions for equal periods of time (12 weeks). Can the authors provide some explanation for this? [I think the problem is in the data in Figure 3B, since there is a great dispersion in the weights of the animals at 5 weeks and very high values can be observed for that age. The evolution of the animals' weights during the 12 weeks of the experiment does not make much sense].
Finally, regarding the experiments that suggest the involvement of mitoKATP channel on the anti-inflammatory and anti-atherosclerotic activities of uridine, it is used 5-hexadecanoate (5-HD) as a very specific channel blocker to suggest its mechanistic participation. However, it has been shown that 5-HD has other targets different from mitoKATP channel (PMID: 12154175) whose blockage could affect the interpretation offered by the authors.
Comments on the Quality of English Language
The use of English is good, but I think the use of some expressions and tenses could be revised for improvement.
Author Response
Reviewer3
In the Abstract the abbreviation ES is used without a prior definition, which makes it difficult to understand what the authors are referring to.
In lines 222, 283, 379, 381 and in Figure 2 μmol is used to refer to a concentration. This should be expressed in the appropriate units.
Reply: Really much thanks for your kind comments. Your opinions are really helpful and constructive to us. We have made corresponding revisions to the article as your suggestions and hope to maintain academic communication with you all the time.
The in vivo and in vitro experiments shown point toward an anti-inflammatory activity of N. brasiliensis ES products and uridine, but, in in vivo experiments using APOE KO mice on high fat diet as a model for atherosclerosis development, the authors focus their interpretation of the results (reduced atherogenesis) on the anti-inflammatory associated effects of the treatment. Nevertheless, it is also shown that uridine treated animals show a serum lipid profile modifications, as well as a lower body weight and reduced fatty liver signs. These facts should be taken into consideration to explain the reduced development of atherosclerotic lesions, as all of them can contribute to the output of the treatment.
I suggest considering the potential effect of the dose of uridine used in the experiment on overall lipid and energy metabolism to give rise, at least partially, to an atheroprotective effect. This could be included in the Discussion section and the manuscript already contains references supporting this point (reference 40).
Reply: Thanks for your advice. We have added relevant description in the discussion.
Regarding the results of the in vivo experiments, it is surprising to me the different behaviour of the animals' body weight throughout the treatment period: in Figure 3B all groups show a reduction in body weight, although it is more pronounced in the mice treated with ES or uridine; in Figure 6B there is an increase of body weight in all groups except those treated with uridine and those treated with 5HD, whose body weight seems to stabilise below the non-treated animals from 4th week, and uridine+5HD treated mice, whose body weight rises until 9th week and then stabilises. It is specially discording among both experiments the fact that for ND and HFD the body weight decreases in one case (Figure 3B) or increases in the other (Figure 6B), assuming that the animals had the same genetic background (APOE KO), the same initial age (5 weeks), and were subjected to similar experimental conditions for equal periods of time (12 weeks). Can the authors provide some explanation for this? [I think the problem is in the data in Figure 3B, since there is a great dispersion in the weights of the animals at 5 weeks and very high values can be observed for that age. The evolution of the animals' weights during the 12 weeks of the experiment does not make much sense.
Reply: Thanks a lot for your suggestion. Regarding the issue of different weight trends between Figure 3B and Figure 6B, it was also strange when we first obtained the results, but we ultimately analyzed some of the reasons from it. A high-fat diet is obviously not the most ideal dietary habit for animals. The mice in Figure 3B were slightly older and have already formed their own dietary habits before modeling. Therefore, there was some resistance during high-fat diet intervention, manifested as unwillingness to eat high-fat feed. However, the mice in Figure 6B, which were slightly younger, did not show such resistance. This might partially explain the reason of the different overall weight trends between two groups. Our evaluation criteria for modeling were mainly based on blood lipids, thus ignoring this weight difference. However, interestingly, the weight differences caused by the intervention treatment group were consistent between the two batches of models.
Finally, regarding the experiments that suggest the involvement of mitoKATP channel on the anti-inflammatory and anti-atherosclerotic activities of uridine, it is used 5-hexadecanoate (5-HD) as a very specific channel blocker to suggest its mechanistic participation. However, it has been shown that 5-HD has other targets different from mitoKATP channel (PMID: 12154175) whose blockage could affect the interpretation offered by the authors.
Reply: Although it cannot be entirely attributed to the role of mitoKATP channel that 5-HD as a specific inhibitor of potassium transport in mitochondria with high selectivity has exhibited a weakened protective effect of uridine to AS in the present study. We should also continue to supplement the detection of mitochondria and exclude the effects of other channels after blocked by 5-HD in the future.
Round 2
Reviewer 2 Report
Comments and Suggestions for Authors
The manuscript has been revised according to the reviewers' comments.
Author Response
The manuscript has been revised according to the reviewers' comments.
Reply: Thanks very much for reviewers' sincere and professional suggestions, which have provided us with great help and inspiration. Based on your comments, we have made corresponding modifications and improvements in the text.
Reviewer 3 Report
Comments and Suggestions for Authors
The authors have responded to the questions raised and have introduced modifications to the manuscript in accordance with the suggestions made.
Author Response
The authors have responded to the questions raised and have introduced modifications to the manuscript in accordance with the suggestions made.
Reply: Sincerely thanks for reviewers' kind comments. It has brought us great benefits communicating with you. We have made modifications to the text based on your suggestions, hoping to provide you with clarification.